# Work: A Social Determinant of Health Worth Capturing

**DOI:** 10.3390/ijerph20021199

**Published:** 2023-01-10

**Authors:** Karla Armenti, Marie H. Sweeney, Cailyn Lingwall, Liu Yang

**Affiliations:** 1NH Occupational Health Surveillance Program, Institute on Disability, University of New Hampshire, Durham, NH 03824, USA; 2National Institute for Occupational Safety and Health, Cincinnati, OH 45226, USA; 3Council of State and Territorial Epidemiologists, Atlanta, GA 30345, USA

**Keywords:** occupational health surveillance, occupation, industry, employment and work, infectious disease surveillance

## Abstract

Work is a recognized social determinant of health. This became most apparent during the COVID-19 pandemic. Workers, particularly those in certain industries and occupations, were at risk due to interaction with the public and close proximity to co-workers. The purpose of this study was to assess how states collected work and employment data on COVID-19 cases, characterizing the need for systematic collection of case-based specific work and employment data, including industry and occupation, of COVID-19 cases. A survey was distributed among state occupational health contacts and epidemiologists in all 50 states to assess current practices in state public health surveillance systems. Twenty-seven states collected some kind of work and employment information from COVID-19 cases. Most states (93%) collected industry and/or occupation information. More than half used text-only fields, a predefined reference or dropdown list, or both. Use of work and employment data included identifying high risk populations, prioritizing vaccination efforts, and assisting with reopening plans. Reported barriers to collecting industry and occupation data were lack of staffing, technology issues, and funding. Scientific understanding of work-related COVID-19 risk requires the systematic, case-based collection of specific work and employment data, including industry and occupation. While this alone does not necessarily indicate a clear workplace exposure, collection of these data elements can help to determine and further prevent workplace outbreaks, thereby ensuring the viability of the nation’s critical infrastructure.

## 1. Introduction

Work is a recognized social determinant of health [1,2] and the way work is structured has an enormous impact on population health. Employment provides substantial benefits to both personal and societal health, and impacts other social determinants of health. However, the work environment may also present substantial risk of worker exposure to physical, chemical, radiological, biological, infectious, ergonomic, and psychological hazards. These hazards may also result in injury, acute or chronic illness, disability, or death, thus interfering with productivity and quality of life. There may also be impacts on income, leisure time, and health insurance availability and coverage (which can also impact health). A person’s industry and/or occupation contribute to the burden and distribution of various chronic illnesses, infectious diseases, and mental health conditions among worker populations. Job characteristics, including wage level or employment stability, impact health even when there is no injury or illness [3]. In public health practice, the industry (i.e., type of business of the employer) and occupation (i.e., the job of the individual) variables are related, and are core socioeconomic variables commonly used as surrogate measures for workplace exposures and hazards to assess the potential contribution of work to the health and wellness of workers.

Non-governmental advisory and professional practice organizations promote and endorse the routine collection of work and employment variables in core public health surveillance systems and all public health surveys [4,5]. Most state vital records departments consistently capture usual industry and occupation on death certificates recorded by funeral directors, coroners or medical examiners, and many states include variables to capture data on work or employment in other records of disease or injury cases. However, standard industry and occupation data are less consistently collected in electronic medical records or for cases of infectious disease.

CDC developed standard case-reporting forms (CRFs) and templates for use by jurisdictional health departments to collect and submit COVID-19 case data as part of case reporting and case notifications sent to CDC. These data are included in the National Notifiable Diseases Surveillance System (NNDSS) [6]. Prior to May 2020, the only data collected on work in the CDC COVID-19 case report form was a yes/no checkbox to indicate whether the case was a healthcare worker. The CDC COVID-19 case report form released 5 May 2020, added questions on categories of healthcare personnel and about workplace exposures in critical infrastructure jobs. The case report forms are kept brief to minimize the burden on health department staff conducting case investigations [7]. While jurisdictions did their best during this early time when the new CRF was released, many struggled to collect even the most basic information about a case due to high case counts. For example, less than 20% of COVID-19 case report forms transferred from jurisdictions to the CDC from March 2020 to March 2021 contained healthcare occupation status [8]. In August 2020, due to the high volume of COVID-19 cases, and to improve completeness of the data that were reported, the CDC recommended limiting notification of COVID-19 cases from state and local health departments to CDC to a core set of data elements, which did not include industry and occupation [9]. While CDC encouraged jurisdictions to return to using the extended case report form and the accompanying COVID-19 message mapping guide (MMG) allow each jurisdiction to map the data elements in their unique systems to a standard message to send a case notification to CDC. The COVID-19 MMG includes all of the data elements found in the current case report form. The data elements in the MMG also have priority designations that guide the jurisdictions as they implement the MMG. (https://ndc.services.cdc.gov/mmgpage/covid-19-message-mapping-guide/ (accessed on 10 October 2022)) ‘once the numbers reduced substantially’, health department capacity to follow up and collect detailed data on every case from providers and patients could not keep up with the volume of COVID-19 case reports [9].

### 1.1. Public Health Implications

The need for collection of core work and employment data was clearly evident during the COVID-19 pandemic. While frontline healthcare workers are at obvious risk of contracting COVID-19, the degree to which non-healthcare workers are at risk due to interactions with the public and close proximity to co-workers is not well-known. In the early part of the COVID-19 pandemic, SARS-CoV-2 infections were differentially distributed by occupation and work setting, with higher prevalence of illness among workers in healthcare, construction, manufacturing, meatpacking, and wholesale trade [10,11,12,13,14]. With the exception of a few studies that examined the prevalence of COVID-19 among specific occupation groups (e.g., meatpacking, food processing, and correctional facilities [15,16,17,18,19]), what resulted was a lack of COVID-19 data describing the general workforce in the U.S., notably essential workers in non-healthcare jobs. In addition, while some individuals could work from home, many others in industries and occupations deemed essential by the Cybersecurity and Infrastructure Security Administration (CISA) and other authorities were required to work outside of the home. This led to increased risk of contact with co-workers, or customers, and may have involved travel using public transportation or company shuttles throughout the pandemic [20,21,22]. 

As the COVID-19 pandemic progressed, the public health community realized that many workers are disproportionately affected by SARS-CoV-2 infections, bringing health inequities in the U.S. into stark relief [23,24,25]. Employment in jobs with higher risk for SARS-CoV-2 infections differed according to demographic characteristics, including race, ethnicity, and nativity. This is due, in part, to occupational segregation, such that people from racial and ethnic minority groups are often employed in occupations that have a higher risk of occupational injuries, illnesses and fatalities [26]. Studies show that Black and Hispanic workers, the groups who have been impacted the most by the pandemic, were more likely to be employed in essential industries as defined by CISA or in occupations with frequent exposure to infections and which required work in close proximity to others [27,28,29,30,31].

A scientific understanding of work-related COVID-19 risk requires the systematic, case-based collection of specific work and employment data, including industry and occupation, of COVID-19 cases. While this alone does not necessarily indicate a clear workplace exposure, collection of these data elements can help to determine and further prevent workplace outbreaks, thereby ensuring the viability of the nation’s critical infrastructure. 

### 1.2. Results from a National Survey on Collection of COVID-19 Case Data

Many jurisdictions have flexibility in operationalizing all but the most basic surveillance recommendations from the CDC and could act to implement the recommendations discussed above for collecting additional information about work and employment. Since the beginning of the pandemic, some states and jurisdictions have reported incidence of COVID-19 by occupation and industry [13,32,33,34,35]. In an effort to better understand when and how states collect work and employment data on COVID-19 cases, the Occupational Health Subcommittee (OHS) of the Council for State and Territorial Epidemiologists (CSTE) developed and conducted a national survey in February/March of 2021. The CSTE Occupational Health Subcommittee, which comprises members representing public health departments and federal partners, as well as academic researchers, provides epidemiology and program planning tools to support state occupational health surveillance capacity. The Subcommittee also serves as a national forum for peer networking, multi-state collaboration, and resource sharing among occupational health epidemiologists, and has been active in leading the efforts within CSTE to inform and increase general understanding of the implications of the pandemic on worker populations. 

## 2. Materials and Methods

A survey was developed and disseminated by the CSTE Occupational Health Subcommittee via email to CSTE state occupational health contacts and state epidemiologists in all 50 states and responses were captured in Qualtrics. The survey included approximately 19 questions regarding state and jurisdictions’ collection of work and employment data for COVID-19 cases in their public health surveillance system.

Responses were downloaded to MSExcel and multiple responses from the same jurisdiction were combined. Analysis was completed using MSExcel and SAS Studio 3.8, and frequency distributions for each question were presented. This activity was reviewed by CDC and was conducted consistent with applicable federal law and CDC policy (See e.g., 45 C.F.R. part 46, 21 C.F.R. part 56; 42 U.S.C. §241(d); 5 U.S.C. §552a; 44 U.S.C. §3501 et seq. This activity was not determined to be research.

## 3. Results

Survey respondents represented state level (28) and one city level (1) health agencies. A total of 23 of the 28 participating states identified that they used either unique state-developed COVID-19 case report forms (16, 57%) or standard CDC case report forms (7, 25%), with 5 states missing this information (18%). States reported using a variety of data collection platforms. More than 40% of the states used a database developed by the jurisdiction, and more than 30% used the National Electronic Disease Surveillance System (NEDSS). Other platforms used included CommCare^®^, EPITrax^®^, ESSENCE, MAVEN^®^, and REDCap^®^. Similarly, a variety of platforms were used to submit data to the Nationally Notifiable Disease Surveillance System (NNDSS) for COVID-19 case notifications to the CDC, including the National Electronic Telecommunications System for Surveillance (NETSS), NBS, HL7 Message Mapping Guides, DCIPHER, and Secure Access Management Services (SAMS). 

Out of the 28 states, 27 reported collecting work and employment information including industry and occupation or other employment related information from COVID-19 cases. For the purposes of this commentary, further analysis was limited to the 27 states that collected work and employment information. Of the 27 states, 24 (89%) reported that work and employment information about COVID-19 cases was collected by state health department, county and/or local jurisdiction staff or outside contractors; one state reported that only outside contractors collected the data. 

One goal of the study was to assess which employment-related data elements were captured for COVID-19 cases, and how states collected that information. Occupation and/or industry information was collected in 25 states (93%) using a variety of ways, with more than half (17) of the states using a separate text-only field, a predefined reference or dropdown list, or a combination of text fields and dropdown lists.

Occupation information was asked slightly more frequently than industry information (Table 1). Of the 24 states who reported collecting occupation, 13 (54%) asked occupation of all cases, while 3 (13%) asked occupation only of healthcare workers. Seven states (29%) stated that they were not able to consistently collect occupation information or only collected the information in prioritized situations (e.g., part of an outbreak) due to capacity limitations. Of the 22 states reporting collecting industry information, 10 (45%) asked industry of all cases, while 3 (14%) only of healthcare workers. Five states (23%) collected industry information incompletely or inconsistently, and two states (10%) stated that the information was collected via employer information.

Eight states coded occupation or industry narrative data using standardized coding structures, among which four states conducted the standardized coding for both occupation and industry. The standardized coding structures used in practice included the Census Industry or Occupation Codes (CIC and COC), the Standard Occupation Classification System (SOC), or the North American Industrial Classification System (NAICS). To code occupation or industry data, three states used the NIOSH Industry and Occupation Computerized Coding System (NIOCCS); one state coded industry to NAICS by linking employer and worker to unemployment insurance data, and another state used a database that associates employers with primary NAICS to identify industry codes.

States reported using industry and occupation data for a variety of purposes. Slightly less than half of the states (13) used the data for outbreak investigations, more than a third (10) used this information to identify high risk populations, about 30% (8) used the information to prioritize vaccination efforts, and more than 20% (6) used the information to assist with reopening plans. States expressed interest in more training related to collecting industry and occupation, specifically online training and learning about real-time industry and occupation coding systems.

Finally, states described barriers to collecting industry and occupation data. Lack of staffing was reported as an issue for most (18, 67%) states, followed by technology issues (9, 33%) and funding (5, 19%). In addition, the states reported staff exhaustion, infrastructure limits, caseload, lack of training, case interview fatigue, convincing decision makers of the need for collection, and moving to population-based measures as additional barriers.

## 4. Limitations

The scope of this study by the CSTE Occupational Health subcommittee was limited to information on states’ data collection and reporting procedures and platforms for COVID-19 case data over a select period of the pandemic. Survey respondents represented state agencies (27), which may not be representative of every state in the U.S. 

## 5. Discussion

The COVID-19 pandemic underscored the need for collection of work and employment information to better understand the risks experienced by various worker groups, particularly those working outside the home, and in close proximity to co-workers, customers and fellow commuters. More importantly, it emphasized the public health need to collect work and employment information consistently and systematically to generate national and jurisdictional-level data to assess the full impact of the pandemic on the health of the US workforce. This survey on collection of COVID-19 case data found that work and employment information is collected and used to some extent, and that states engage a variety of organizational levels to collect work and employment data. Critical barriers to collecting industry and occupation highlight the need for support of additional staff at the jurisdictional level to reduce the burden on public health responders and contact tracers, particularly when the volume of cases rises, as well as the need for data collection platforms to standardize and automate the collection and coding of industry and occupation data elements from cases. Other barriers include staff exhaustion, caseload, lack of training, and convincing decision makers of the need for collection of the industry and occupation variables. 

Without the availability of systematically collected national case surveillance data that includes work and employment variables, there is a gap in the understanding of the risk of occupational exposure to COVID-19 and its impact on the full spectrum of worker groups or the magnitude of that risk. Challenges to effective surveillance of COVID-19 cases highlight how modernization of the US public health surveillance systems could improve the ability to protect the public (including workers) from all diseases and conditions of public health significance [36] In addressing these challenges, the opportunity exists to enhance these surveillance systems to include standardized collection of both industry and occupation variables.

Regarding the collection of industry and occupation variables, the National Institute for Occupational Safety and Health (NIOSH) has developed and tested standard questions to collect industry and occupation on surveys and in data collections for national surveillance through NNDSS, as well as on other health forms, notably the Behavioral Risk Factor Surveillance System (BRFSS) [37]. There are also standard questions on official US death certificates. These models have been applied to the collection of notifiable diseases, including COVID-19 (through the COVID Message Mapping Guide).

In order to ease the coding burden on jurisdictions collecting industry and occupation variables for disease surveillance, NIOSH encouraged use of the free web-based application NIOCCS, or the NIOSH Industry and Occupation Computerized Coding System, to assist jurisdictions in coding industry and occupation text to established schema such as the North American Industrial Classification System (NAICS), Standard Occupation Classification (SOC) and the Census Industry and Occupation Classification System [38]. NIOCCS is used by NIOSH and the states to code industry and occupation data recorded on most US death certificates, the BRFSS, NNDSS, and a variety of other surveillance systems and surveys. Coding industry and occupation to established, standardized categories allows for comparison of data across jurisdictions and studies.

Finally, in support of modernized national public health surveillance through NNDSS, NIOSH, in partnership with the National Center for Immunization and Respiratory Diseases (NCIRD), recognized the need to increase the capacity of state, local and territorial jurisdictions to capture harmonized and standardized work-related data elements. Funding was provided through the Coronavirus Aid, Relief, and Economic Security (CARES) Act and the Coronavirus Response and Relief Supplement Appropriations Act to enhance collection and use of industry and occupation surveillance data for COVID-19 and other vaccine-preventable and respiratory conditions. This CDC support has resulted in several jurisdictions codifying the collection of work-related information, adding industry and occupation to infectious conditions under surveillance, incorporating coding of work-related information into standard protocols, and implementing processes to automatically code industry and occupation data during case interviews. NIOSH also has a variety of resources to help jurisdictions with training needs of staff assigned to case investigation, contact tracing, and data collection for public health surveillance [39].

## 6. Conclusions

Awareness of the prevalence of disease outcomes by industry and occupation can assist jurisdictions in identifying disease clusters or outbreaks, determining the magnitude of a problem within worker groups, and prioritizing industries and occupations for the development of prevention measures. Consistent collection of work and employment data, including industry and occupation as standard demographic data elements, would give jurisdictions the ability to provide more comprehensive, and more representative data on the distribution of injury and all illnesses by worker groups. Inclusion of industry and occupation data helps identify essential industries and occupations most impacted by a widespread infection and substantial health inequities. Prioritization efforts to address these health inequities may create change in many systems. Jurisdictions could use the data to design better health promotion programs for workers. Many of these programs are employment-based, including cancer screenings, vaccination uptake, and prevention of chronic conditions. Finally, jurisdictions can disseminate findings of studies or case investigations to engage employers, who bear a large share of health care costs, as active partners in ensuring a safe environment for their workers. 

Findings from this survey have provided an opportunity to reflect on what we have learned during the COVID-19 pandemic to improve occupational health and exposure surveillance efforts going forward. Highlighting the importance of collecting work and employment data, including industry and occupation, is timely given CDC’s current efforts to enhance data systems and its review of response activities. These data can be used to assist jurisdictions at all levels in future response activities. With work now being a recognized social determinant of health, use of work and employment information, including industry and occupation, is a critical component of core public health surveillance systems. Collection of these variables is important both for routine surveillance activities and crisis responses. 

## Figures and Tables

**Table 1 ijerph-20-01199-t001:** Number of states collecting employment-related data (*n* = 27).

	Number of States (%)
Employer name	21 (78%)
Employer address	16 (59%)
Is the case a HCW?	24 (89%)
Occupation	24 (89%)
Industry	22 (81%)
Was the case at work in the 14 days before diagnosis or symptoms?	20 (74%)

## Data Availability

Not applicable.

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
