# Peer review of "Work: A Social Determinant of Health Worth Capturing"

_ijerph, 2023, doi:10.3390/ijerph20021199_

Round 1

Reviewer 1 Report

During the COVID-19 pandemic, a lot of new problems have appeared that must be solved by governments with the help of scientists. This article is devoted to one of these problems. However, the article is not so much scientific as it is analytical in nature. Actually, the authors attempt to assess how effective the system of public health surveillance of a separate state is.

To make the article look more scientific, I can suggest the following improvements:

1) In the Introduction, it is appropriate to justify the problem not only from the standpoint of the government and practice, but primarily by referring to scientific publications with appropriate arguments and identifying the unresolved part of the problem. It is appropriate to cite the cases of different countries, since the pandemic has covered the whole world.

Besides, please remove identical phrases that contain the text of the article and the abstract, for example "Work is a recognized social determinant of health".

Moreover, state the purpose of this study more clearly.

2) In the Materials and Methods, it is necessary to strengthen the description of the methods used during the research and justify why these methods are the most appropriate for this study. Besides, missing is a table describing the data, including the most important details of the sample.

3) The results are presented rather sparsely. In order to understand them, it is better to add a picture or a table, which reveals their essence and benefits for the reader in more detail.

4) The discussion can be strengthened by comparing the results of the study with similar results from other countries and from previous studies.

Author Response

  • In the Introduction, it is appropriate to justify the problem not only from the standpoint of the government and practice, but primarily by referring to scientific publications with appropriate arguments and identifying the unresolved part of the problem. It is appropriate to cite the cases of different countries, since the pandemic has covered the whole world.

Response: Thank you for your thoughts. We provided citations to many articles to support the current literature. We did not feel it appropriate to expand to the experiences of other countries because the issues were specific to the United States. Furthermore, this article is a commentary and therefore does not follow the protocols for a traditional research study. 

Besides, please remove identical phrases that contain the text of the article and the abstract, for example "Work is a recognized social determinant of health".

Response: Generally,  an abstract repeats (and summarizes) what is in the body of the article. No change.

Moreover, state the purpose of this study more clearly. 

Response:  We have made a minor revision to the Abstract stating the purpose of the study.

2) In the Materials and Methods, it is necessary to strengthen the description of the methods used during the research and justify why these methods are the most appropriate for this study. Besides, missing is a table describing the data, including the most important details of the sample.

Response: Our article is a commentary and therefore does not follow the protocols for a traditional research study.  We state in the article that the study was not deemed to be research but is a convenience sample as noted in the commentary.

3) The results are presented rather sparsely. In order to understand them, it is better to add a picture or a table, which reveals their essence and benefits for the reader in more detail.

Response:  There is one table that we thought was the most important to show.  With such a small sample, it would not be meaningful to produce more tables or graphs.

4) The discussion can be strengthened by comparing the results of the study with similar results from other countries and from previous studies.

Response: Same comment as # 1 and 2 and our focus was on data collection in the US, not in other countries.

Reviewer 2 Report

The draft assesses the pros and cons of current practices in state public health surveillance systems by collecting work and employment data on cases of Covid-19, it can play a good role in guiding and drawing lessons for the formulation of epidemic prevention and control strategies in the future, but its deficiency is that the analysis of the relevant data of different industries, occupations and populations is not careful enough.

1. What are the specific impacts of covid-19 on different industries and populations? To what extent?

2. It is suggested to clarify what should be covered by the unified and standardized data and list them one by one.

3. Recommend the listing of harmonized occupational and Industry Data and the identification of their advantages and uses.

Author Response

The draft assesses the pros and cons of current practices in state public health surveillance systems by collecting work and employment data on cases of Covid-19, it can play a good role in guiding and drawing lessons for the formulation of epidemic prevention and control strategies in the future, but its deficiency is that the analysis of the relevant data of different industries, occupations and populations is not careful enough.

Response: Thank you for your comment. This article is a commentary and therefore does not follow the protocols for a traditional research study. We had no data on industries or occupations but we did cite articles focused on this topic.

  1. What are the specific impacts of covid-19 on different industries and populations? To what extent?

Response: We provide many citations highlighting this.

  1. It is suggested to clarify what should be covered by the unified and standardized data and list them one by one. And 3. Recommend the listing of harmonized occupational and Industry Data and the identification of their advantages and uses.

Response: We recommend the collection of work and employment variables to be added to the current system.  Our goal is not to revise the current unified or harmonized data systems that states use to collect information on reportable cases of infectious diseases, though a more consistent data collection system could be very useful. However, this is not the focus of the commentary.